# Detection of Wheel Polygonization Based on Wayside Monitoring and Artificial Intelligence

**DOI:** 10.3390/s23042188

**Published:** 2023-02-15

**Authors:** António Guedes, Ruben Silva, Diogo Ribeiro, Cecília Vale, Araliya Mosleh, Pedro Montenegro, Andreia Meixedo

**Affiliations:** 1CONSTRUCT-LESE, School of Engineering, Polytechnic of Porto, 4200-465 Porto, Portugal; 2CONSTRUCT-LESE, Faculty of Engineering, University of Porto, 4200-465 Porto, Portugal

**Keywords:** wheelset, dynamic analysis, wheel polygonization, wayside monitoring system, automatic wheel defect detection

## Abstract

This research presents an approach based on artificial intelligence techniques for wheel polygonization detection. The proposed methodology is tested with dynamic responses induced on the track by passing a Laagrss-type rail vehicle. The dynamic response is attained considering the application of a train-track interaction model that simulates the passage of the train over a set of accelerometers installed on the rail and sleepers. This study, which considers an unsupervised methodology, aims to compare the performance of two feature extraction techniques, namely the Autoregressive Exogenous (ARX) model and Continuous Wavelets Transform (CWT). The extracted features are then submitted to data normalization considering the Principal Component Analysis (PCA) applied to suppress environmental and operational effects. Next to data normalization, data fusion using Mahalanobis distance is performed to enhance the sensitivity to the recognition of defective wheels. Finally, an outlier analysis is employed to distinguish a healthy wheel from a defective one. Moreover, sensitivity analysis is performed to analyze the influence of the number of sensors and their location on the accuracy of the wheel defect detection system.

## 1. Introduction

Since its emergence, rail transport has been preponderant in developing societies’ living conditions. It is an effective and efficient transportation system for both passengers and goods, for small and large amounts of cargo, as well as for short and long distances [1]. In an increasingly interconnected world, there is a need to continuously update and expand these systems. Moreover, the significant increase in traffic speeds imposes a greater dynamic load on structures [2,3,4]. In this sense, it is essential to implement methodologies for monitoring existing infrastructures and vehicles to assess current safety conditions and limitations [5,6], as well as for the detection of failures or defects at an early stage and, consequently, minimizing damage [5,7,8].

Condition monitoring systems are commonly referred to as wayside and on-board systems, distinguishing them based on the location of the measuring devices, both on the track and the vehicle. Installing sensors on in-use vehicles is useful for monitoring track component failures. For vehicle defects, it is common to adopt trackside embedded systems [9]. Vehicle damage detection methodologies are mainly focused on detecting damage on wheelsets and bogie components, as they have a greater impact on vehicle performance and represent the highest percentage of maintenance costs [6].

The application of sensors on the track allows the extraction of significant amounts of data referring to all the operating vehicles, thus enabling the methodology to monitor various vehicles with a reduced number of sensors. Langnebäck [1] pointed out that systems applied to vehicles for detecting damage on the vehicle components are quite expensive, since sensors are required in all vehicles. Therefore, this method becomes economically infeasible since the costs would be higher than the cost of maintenance or repair caused by the damage to be identified. In addition, the substantial number of existing vehicles represents a significant challenge in terms of the organization and maintenance of the technology for detecting damage in each of them. In summary, despite the high precision for monitoring defective wheels, these on-board systems require a high installation cost to study a wider range of vehicles.

In a wayside monitoring system, the wheel-rail contact interface plays a significant role. At this point, the dynamic forces of wheel-rail interaction are established, which allows the acquisition of relevant information about various factors inherent to the traffic conditions [1]. Although these components represent a high maintenance cost, serious accidents with high economic losses and possible human injuries can be avoided. Regarding railway safety, wheel defects are one of the main causes of accidents [10,11], causing interruptions in the normal operation of railway transport systems and considerably reducing their reliability. Furthermore, these defects promote negative effects on both the vehicle and track, increasing maintenance costs. 

Furthermore, wheels deteriorate over time due to wear and fatigue, which may develop various defects such as polygonization, flats and shelling, altering the wheel-rail contact characteristics and, consequently, causing greater impact forces [12]. Although a polygonized wheel generates fewer effects on the track compared to a wheel flat [13], this defect may cause serious damage to the track and the vehicle due to the excitation promoted by this phenomenon. It also increases noise emissions inside and outside the vehicle, and decreases comfort levels and railway safety [9]. Moreover, depending on the characteristics of the vehicles and the number of wavelengths of the polygonal defect of the wheel, the vibration frequencies caused by the defect can induce resonance effects in the wheelsets increasing the vibration on these components [13,14]. In the last two decades, polygonization in wheels has been the subject of great interest in the scientific community. Johansson and Andersson [15] used a mathematical model to predict polygonal wear on train wheels with harmonics of 1–20. In the research study by Cai et al. [16], the mechanism of polygon formation on metro wheels was investigated based on experimental measurements and numerical simulations. Further, in the study by Peng [17], the mechanisms of OOR formation on wheels were presented considering different wear prediction models. 

Considering recent technological advances, it is now possible to develop and implement automatic methodologies for detecting and identifying damage in different systems [15,16,17] for railway applications. With the application of these intelligent techniques, it is not intended to completely replace the visual inspections already used for many decades, but rather, the creation of a compatible procedure. Traditional visual inspection and control techniques are expensive, occur at intervals, and are prone to errors, it is important to motivate the widespread use of Structural Integrity Monitoring (SIE) systems, especially in recent infrastructures or in systems of high value to society, as in the case of railway systems [18]. Thus, it is essential to develop automatic methodologies for detecting different types of wheel defects, such as polygonization wheels.

Despite the widespread research on railway defect detection, to the knowledge of the authors, the reported literature on automatic wheel polygonization identification has been limited so far. Therefore, taking advantage of artificial intelligence (AI) methodologies, it is important to improve safety and reduce operating costs by identifying a defective wheel at an early stage. The detection of polygonized wheels is largely carried out using accelerometers mounted on the vehicle. The parametric power spectral estimation and adaptive chirp mode decomposition techniques are common approaches used with satisfactory detection results, but with flaws in polygonization detection in the early stages [19,20,21].

On the other hand, previous studies have shown very good results using wayside track monitoring and automatic damage classification methodologies based on machine learning in detecting defects in railway vehicle wheels, namely wheel flat detections [16,22]. This type of methodology usually presents the following steps: (i) feature extraction using data from accelerations or strains measured on the track, (ii) feature modeling with the application of normalization techniques to reduce the effects of environmental and operational variations, (iii) data fusion to merge different features and/or sensors information, and (iv) features discrimination to classify the defects. The studies carried out concerning the automatic detection of wheel imperfections are mostly dedicated to identifying singular irregularities on the wheel surface (i.e., wheel flat). In this context, it is important that monitoring systems can cover the detection and classification of a more comprehensive number of damages, namely the ones that may occur along the entire perimeter of the wheel, such as polygonization.

Several methods have been used for feature extraction, which converts time-series data into more condensed information, allowing damage to be more easily observed. Symbolic data, continuous wavelet transform (CWT) [15,16,17], principal component analysis (PCA) [23,24], and autoregressive models [15,16,17] are examples of effective techniques for extracting damage-sensitive features for both static and dynamic monitoring. In time-domain applications where the measured quantity is acceleration, autoregressive models have been widely used due to their easy computational implementation and solely relying on the system response, with the model parameters reflecting the properties of the system regardless of the sources of excitations. CWT and PCA have been developed over decades in different applications, in both time and frequency domains with various sensors. Despite this, they present some limitations, PCA does not consider the non-linearity of the data and CWT adds excess redundancy to the data, and it has a high computational cost, so normally, they are used in offline analysis applications.

For feature fusion, several algorithms, including neighborhood-preserving embedding (NPE) [25], neural networks [26], Mahalanobis distance [27], manifold learning methods [28], and kernel-based methods [29], have been recently employed. This data fusion aims to reduce the volume of the extracted features by persevering the most relevant information. Due to its simplicity and ability to reduce multivariate data, Mahalanobis distance is widely used [30].

Recent approaches based on machine learning have been applied for feature classification to differentiate a healthy wheel from a defective one [9,31]. Typically, two different approaches are used: (i) unsupervised methods, in which models are trained using labeled data under the supervision of training data, such as k-means [15], self-organizing maps (SOM) [32], and cluster analysis; or (ii) supervised methods, in which models are not supervised using training datasets, such as Naive Bayes classifiers [33] and k-Nearest Neighbor classifiers [34].

This work aims: (i) to develop an unsupervised methodology to identify polygonized wheels; (ii) to evaluate the performance of different feature extraction techniques concerning their sensitivity to the detection of wheel polygonization, specifically those based on autoregressive exogenous models (ARX) and continuous wavelet transforms (CWT); (iii) to analyze the influence of the number of sensors and their location on the accuracy of the wheel defect detection system. The unsupervised methodology to identify polygonized wheels consists of five phases: (i) data acquisition, (ii) feature extraction, (iii) feature modeling, (iv) data fusion, and (v) feature discrimination.

The acquired track responses are influenced by parameters such as speed, irregularities, and load types. In this sense, different simulations of realistic scenarios of dynamic vehicle-track interaction were considered in view of a simulation based on real operational conditions. On the other hand, to overcome some unpredictable oscillations in the measured responses and consequent failures in damage detection, some options can be taken. In more complex structures, such as bridges and viaducts, there should be a greater susceptibility to damage when compared to areas of track sections on the foundation ground. If the structures are damaged, they may camouflage or cause false warnings. In a practical context, the use of sensors in an area of the track less prone to damage and where vibration frequencies are more stable provides a more favorable environment for the success of the methodology. Aligned on these assumptions, the location of the sensors was chosen on a straight stretch of the track less subject to large variations in vibration frequencies when traffic passes.

After defining the AI approach for identifying wheel polygonization, two distinguished feature extraction methods, namely ARX and CWT, are compared. To avoid false alarm situations caused by environmental and operational effects, the latent variable method Principal Component Analysis (PCA) is applied. To reduce the volume of extracted data while retaining the most relevant information, as well as improve the ability to characterize the measured phenomenon, the data are merged using the Mahalanobis distance. As a final step, an outlier analysis is employed to automatically distinguish a healthy wheel from a defective one.

It should be highlighted that wheel flat identification was developed by Mosleh et al. [16,22], who demonstrated that the methodology is able to identify wheel flats. The validation of the proposed AI-based methodology herein, considering a polygonal wheel, is a clear step forward in terms of the proposed methodology’s effectiveness, allowing for a complete implementation for real-world application.

## 2. Proposed Methodology for Automatic Wheel Polygonization Detection

This work presents a data-driven unsupervised method for automatically detecting wheel polygonization in five steps, as presented in Figure 1 and described below [15,16,22,35]. 


(i)Data acquisition from accelerometers.(ii)Feature extraction from multiple sensor signals, by continuous wavelet transforms (CWT) or autoregressive exogenous (ARX) models, that are separately implemented, and their performances are compared; in this step, a time record is transformed into features that are damage sensitive.(iii)Feature modeling based on the Principal Component Analysis (PCA) technique that not only normalizes the extracted features but also increases their sensitivity to damage.(iv)Data fusion by applying Mahalanobis distances (MDs) to the features to enhance their sensitivity. By using the Mahalanobis distance, the features from each sensor can be effectively merged in the first stage. Moreover, a new Mahalanobis distance is applied again in the second stage to integrate all sensor information. The outcome of this step is a damage indicator (DI), for each train passage.(v)Feature discrimination is performed by statistical analysis to determine whether a DI identifies a healthy or a defective wheel; a statistical confidence boundary (CB) is estimated based on an inverse cumulative distribution function.


### 2.1. Feature Extraction

In this study, the features are extracted based on autoregressive models with exogenous inputs (ARX) and continuous wavelet transform (CWT) models. The performance of each technique is studied and compared to obtain the result most sensitive to wheel polygonization.

#### 2.1.1. ARX Model

Yang and Makis [36] present a method based on ARX parameters capable of detecting the occurrence of gearbox failures. The ARX autoregressive model considers the predictive behavior of the system at a specific location, considering its history at the same point and the predictive response at other measurement locations, being defined by the following equation:(1)$$x_{j}=\overset{m_{a}}{\underset{i=1}{\sum }}a_{i}x_{j-i}+\overset{m_{b}}{\underset{k=1}{\sum }}b_{k}y_{j-k}+\varepsilon_{j}$$
where $x_{j}$, $y_{j}$, and $\varepsilon_{j}$ are the output, input, as well as error term of the model at the signal value j. Moreover, $a_{i}$, $b_{k}$, $m_{a}$, and $m_{b}$ are the parameters and the orders of the output and input data. 

Initially, it is difficult to determine the most appropriate ARX model order. The key issue is that higher-order models could better match the data, but they may not generalize to other datasets. Otherwise, a low-order model is unlikely to capture the underlying physical system response. Several authors have noted the difficulty of defining a unique order for ARX models. The Akaike Information Criterion (AIC) is reported to be one of the most effective optimization techniques [18,37]. Based on the exchange between fit accuracy and the number of estimated parameters, AIC measures the goodness-of-fit of an estimated statistical model that is determined by the following function:(2)$$AIC=N_{t}\ln\varepsilon +2N_{p}\ldots ,\varepsilon =\frac{SSR}{N_{t}}$$
where $N_{p}\;$ and $N_{t}$ are the number of estimated data points and the number of predicted parameters, respectively, and ε is the average sum-of-square residual (SSR) errors. To find out an optimum order, one should examine a wide range of orders and choose the order number with the minimum AIC value.

#### 2.1.2. CWT

Studies conducted by Li et al. [38] showed good results in gearbox fault detection. CWT produces a two-dimensional set of coefficients from the analyzed signal. Although the Fourier transforms typically use sine waves as the basis for decomposition, other functions can be selected for the wavelet shape due to signal properties. Two parameters define wavelet analysis: scale and translation. The CWT (T) of the signal x(t) is determined as follows [39]:(3)$$T\left(a,b\right)=\frac{1}{\sqrt{a}}\int_{-\infty }^{\infty }x\left(t\right)\psi^{*}\left(\frac{t-b}{a}\right)dt$$
where a and *b* are the scale and translation parameters, and $\psi^{*}$ is the complex conjugate of the mother wavelet T  that involves a continuous function in both time and frequency domains.

After the implementation of a CWT on the evaluated signals, a PCA is performed followed by the extraction of four statistical parameters, resulting in significant data compression [17,40]. Based on the *p*-by-*q* matrix *X* containing the extracted CWT features from the original signal evaluated by each sensor, where *p* is the measurement points number and *q* is the wavelets coefficients number, the principal components can be determined using the following equation:(4)$$S_{s}=XT$$
where $S_{s}$ represents the scores matrix, and *T* is a *q*-by-*q* orthonormal linear transformation matrix. In addition, four statistical parameters are extracted from the scores, $S_{s}$, including the root mean square (RMS), standard deviation, skewness, and kurtosis. Thus, a matrix *p*-by-*q* is transformed into a *k*-by-*q* matrix, where *k* equals 4. Consequently, a total of *k × q* features are extracted for each sensor and each train passage.

### 2.2. Feature Modeling

The feature modeling process aims to remove the changes in features resulting from environmental and operational variations (EOVs). For example, in damage detection, it is challenging to remove EOVs effects from dynamic properties to obtain features that are mostly sensitive to damage [16]. Therefore, implementing a latent variable method, namely the PCA, may efficiently reduce EOVs influences, without requiring the direct measurement of those effects. According to Bro and Smilde [41], PCA is a powerful and versatile method capable of providing an overview of complex multivariate data. PCA can be used to find and quantify patterns, generate new hypotheses, detect outliers, or find among sample data in order to create clusters.

By considering a *m*-by-*f* matrix X, taking into account features extracted from the dynamic responses, where *m* represents the train passages and *f* represents the number of features extracted from sensors (i.e., *k × q*), a transformation to another set of *f* features, $S_{s}$, is obtained by considering Equation (4). Considering the baseline condition, the covariance matrix (*C*) of the features is related to the covariance matrix of the scores Λ, as follows:(5)$$C=T\Lambda T^{T}$$
where, Λ and T are matrices calculated by the singular value decomposition of the covariance matrix C. The columns of T are the eigenvectors and the diagonal matrix Λ, including the eigenvalues of the matrix C in descending order. Thus, the eigenvalues stored in Λ are the variances of the components of $S_{s}$ and provide the relative importance of each principal component in the entire dataset variation [30].

It is possible to divide matrix Λ into two matrices with the first *n* eigenvalues and the remaining *f-n* eigenvalues. Determining the number of *n*-components is one of the challenges in multivariate data interpretation. In this research study, *n*-components are discarded when the cumulative percentage of variance reaches 80% [15,35,42,43]. Once *n* has been selected, Equation (4) can be used to calculate the f-n components of matrix S. The transformation matrix $\hat{T}$ is constructed by the remaining *f-n* columns of matrix  T. Using the following function, the *f-n* components can be remapped to the main space:(6)$$F_{PCA}=X\hat{T}{\hat{T}}^{T}$$
where, $F_{PCA}$ is *m*-by-*f* matrix of PCA features which is less sensitive to EOVs. Every sensor is subjected to the abovementioned process.

### 2.3. Data Fusion

Mahalanobis Distance (MD) is a tool that promotes the improvement of the damage index definition, is easy to implement and fast to process [44]. For each simulation, the MD is considered to calculate a damage index (DI). Several damage identification studies have used this criterion because of its simplicity and ability to reduce multivariate data to one DI [15,24,45]. The MD is a well-known distance metric because it measures the distance between two points in a feature space containing two or more variables [37]. Furthermore, it considers correlations between variables and does not depend on the scale of the features. It can merge features from multiple sensors by performing data fusion. Mahalanobis distance (namely here as DI) is calculated by the following function:(7)$$DI=\sqrt{\left(x_{i}-\bar{x}\right)\cdot C^{-1}\cdot {\left(x_{i}-\bar{x}\right)}^{T}}$$
in which, the inverse covariance matrix of the baseline simulation is defined by $C^{-1}$, and $\bar{x}\;$ is a mean vector of the features from the baseline simulation. The test vector of *f* features representing the potential damage is defined by $x_{i}$. The covariance matrix and mean vector intend to incorporate the baseline simulation. Based on the data derived from the damaged system, a new observation would be far from the average of normal conditions. Mahalanobis distances are calculated for each sensor and simulation. A matrix containing *m* Mahalanobis distances for *i* sensors is obtained from the above calculations.

### 2.4. Feature Discrimination

To automatically monitor train wheel conditions, the methodology performs outlier analysis. In Meixedo et al. [15] and Mosleh et al. [16], this tool proved to be very efficient in classifying the data as being related to a scenario with or without damage. Mahalanobis squared distance is generally approximated by chi-squared distributions in *n*-dimensional space. A Gaussian distribution can be used to approximate the Mahalanobis distance, and statistical thresholds can be used to analyze outliers. By considering a mean value $\bar{\mu }\;$ and standard deviation σ of the baseline feature vector, as well as the significance level α, it is possible to estimate a confidence boundary (CB) for detecting a DI including an outlier using the Gaussian inverse cumulative distribution function (ICDF):(8)$$CB=invF_{x}\left(1-\alpha \right)$$
where
(9)F(x| μ¯,σ)=1σ2π∫−αxe−12(x−μ=σ)2dy,    c/ xϵℝ
consequently, when *DI* is equal to or higher than *CB,* a feature is considered to be an outlier.

## 3. Numerical Models

### 3.1. Train

In the present research, the Laagrss freight train composed of five wagons is studied. According to the UIC classification, it can reach a top speed of 120 km/h [46]. Each wagon has a tare weight of 27 t and can carry loads up to 52 t. A double freight wagon is shown in Figure 2.

ANSYS^®^ (2018) [47] was employed to develop a 3D multibody dynamic model that simulates suspensions in all directions with spring-damper elements and mass point elements to represent the effect of mass and inertia at the center of gravity for each wagon component. A rigid beam element was also used to connect the above-mentioned components. The numerical model of the vehicle is presented in Figure 3 and the mechanical and geometric properties of the vehicle are detailed in Table 1. More details regarding the numerical model of the freight wagons are provided by Bragança et al. [48].

### 3.2. Track

The numerical model of the track was carried out in the ANSYS^®^ [47] finite element program, developed by Montenegro et al. [49]. The model is based on a multi-layer scheme, simulating the ballast, sleepers, and rails, as shown in Figure 4. The railpads, resting on the sleepers under the rail, are simulated as spring elements connecting the sleepers and rail. The mass of the ballast is represented by elements of discrete mass points, while the rails and sleepers are modeled using beam elements, adopting the appropriate material properties for each one. Finally, spring-dashpot elements are also incorporated to consider foundation flexibility. Table 2 presents the properties of the track model.

### 3.3. Track Irregularities

There are imperfections in rails in real-track conditions. Even though these irregularities are very small, their effects on wheel-rail contact cannot be neglected [50]. Therefore, rail unevenness profiles are generated for wavelengths between 1 m and 75 m, corresponding to wavelength intervals D1 and D2 defined by the European Standard EN 13848-2 [51]. Therefore, based on actual data, PSD curves are developed to generate artificial unevenness profiles. More information regarding the generation of unevenness profiles was provided by Mosleh et al. [52]. Figure 5 shows four unevenness profiles of the rail corresponding to the lateral and vertical irregularities on the right rail.

### 3.4. Wheel Polygonization Profiles

In real conditions, the wheels are not entirely smooth and have imperfections. Despite their small size, these out-of-roundness irregularities can cause extreme variations in the wheel–rail contact forces, producing vibrations on the train and track components. A periodic radial tread irregularity around the wheel circumference characterizes the wheel polygonization. According to Peng [53], for wheel polygonization, the irregularity amplitudes should be higher than 0.2 mm. Inherent to polygonization, some parameters can geometrically characterize the phenomenon, such as the varying wavelengths (λ) in the function of the harmonic order (θ) and wheel radius (*R_w_*), as defined by the following function:(10)$$\lambda =\frac{2\pi R_{w}}{\theta },\;\;\;\theta =1,2,3\ldots ,n$$

Two groups of polygonal wheel profiles are generated for the numerical simulations, one for undamaged scenarios and another for damaged scenarios, based on real wheel measurements, the irregularity amplitude spectra of which are presented in Figure 6.

For the first group of profiles, the measurement data obtained by Johansson et al. [54] for new wheels with wavelengths comprising 20 harmonics were used (Figure 6a). The wheels have an initial level of polygonization that includes a wide spectrum of wavelengths between 0.135–2.7 m, being the upper limit of the perimeter of the wheel. For the second group of profiles, the measurement values for four wheels with polygonal damage obtained by Cai et al. [55] were considered (Figure 6b). The wheel profiles are characterized by the wavelengths in the first 30 harmonics, with the 6th to 8th harmonic orders being dominant.

Then, different irregularity wheel profiles are generated based on the sum of sine functions (H = 30) as follows: (11)$$w\left(x_{w}\right)=\overset{H}{\underset{\theta =1}{\sum }}A_{\theta }\sin\left(\frac{2\pi }{\lambda }x_{w}+\psi_{\theta }\right)$$
where $A_{\theta }$ is the amplitude of the sine function for each wavelength, which is calculated by the following function:(12)$$A_{\theta }=\sqrt{2}\cdot 10^{\frac{L_{w}}{20}}\cdot w_{ref}\;\;$$
where, as $w_{ref}=1\;\mu m.$ The wheel irregularity level (*Lw*) values are selected based on the irregularity spectrums of Figure 6, for both cases: initial polygonization (Figure 6a) or polygonal damage (Figure 6b). Considering phase angles ($\psi_{\theta }$) to the sine functions that are uniformly and randomly distributed between 0 and 2π, several wheel irregularities are generated by each spectrum. 

In the case of polygonal damage, simulating different amplitudes are applied at different scale factors for the irregularity profiles generated with Equation (11) in order to obtain two different damage severities in the function of amplitude ranges, one between 0.2 mm and 0.6 mm (a1) and the other between 0.8 mm and 1.2 mm (a2). Figure 7 and Figure 8 show examples of wheel profiles generated for the case of initial polygonization and polygonal damage profiles, respectively. By observing these two figures, the difference in irregularities amplitudes between an initial state of a polygonal wheel with amplitudes around 0.035 mm and a damaged polygonal wheel with amplitudes between 0.2 mm and 1.2 mm can be observed.

### 3.5. Train-Track Dynamic Interaction

Simulations of the dynamic interactions between train and track were carried out using the in-house software VSI—Vehicle-Structure Interaction Analysis, which was validated and described in detail in the authors’ previous publication [56] and was used in various of applications [49,57]. In this model, the train is coupled to the track using a 3D wheel-rail contact model, using Hertzian theory to calculate the normal contact forces and the USETAB routine to calculate the tangential forces resulting from the rolling friction creep phenomenon. As a numerical tool, this software uses MATLAB^®^ [58] to import the structural matrices of both vehicles and track previously modeled in finite element (FE) software. It is important to note that both subsystems are initially separately modeled in ANSYS^®^ [47] (as explained in the previous sections), and the VSI software integrates their models via a fully coupled approach (see [56]). The full explanation of the properties of track and train, including more details regarding train-track interaction, can be found in the authors’ previous publications [59,60]. A graphical representation of the numerical model is shown in Figure 9. Note that the first wheel of the first wagon on the right side is considered a defective wheel (marked in red color).

## 4. Simulation of Baseline and Damaged Scenarios

A series of accelerometers are mounted along the track to detect a polygonized wheel. According to the results demonstrated in Mosleh et al. [16,22], the location of the most promising sensors for obtaining good results is on the rail. However, the possibility of placing the sensors on the sleepers was also analyzed, since their installation is much easier. Using a virtual wayside monitoring system, acceleration measurements were evaluated from 16 positions, as shown in Figure 10. In this figure, the numbers 1-to-4 and 5-to-8 represent the position of the measurement points mounted on the rail between two sleepers on the right and left sides, respectively, while the accelerations over the sleepers are measured by sensors located on positions 9-to-12 and 13-to-16. A virtual simulation of undamaged and damaged wheel scenarios was conducted to test and validate the automatic wheel polygonized detection approach presented in this study. Once validated, this method can be applied to real experimental data for different types of trains and considering different wheel defect conditions. 

Based on several speeds, loading schemes, and unevenness profiles of the rail, Table 3 summarizes the simulations of the undamaged and damaged scenarios. Baseline scenarios consider vehicle speeds between 40 and 120 km/h, and four unevenness profiles are considered for rail irregularities (Figure 5). A total of six types of loading schemes are considered: (i) a fully loaded train, (ii) a half-loaded train, (iii) an empty train, and trains with an unbalanced load in the transversal and longitudinal directions, namely (iv) UNB1, (v) UNB2, and (vi) UNB3. In accordance with UIC loading guidelines [46], different unbalanced loading schemes can be applied to the wagon model where the cargo center of gravity is longitudinally and transversally offset. For baseline scenarios, 113 simulations are conducted with healthy wheel profiles and 30 simulations are carried out considering wheels with initial polygonal profiles. Therefore, in addition to 113 analyses with healthy wheels, 30 scenarios are considered with three semi-imperfection cases, including: (i) the right wheel on the front wheelset of the first wagon, (ii) the left wheel of the rear wheelset of the third wagon; (iii) the right wheel of the rear wheelset of the fifth wagon. Furthermore, 30 damaged scenarios with several combinations of defect amplitude are implemented at the front wheel on the right side of the first wagon as a polygonized wheel. These scenarios are simulated with a freight train circulating at 80 km/h.

Acceleration signals are evaluated at a sampling frequency of 10 kHz for both baseline and damaged scenarios. With a cut-off frequency of 500 Hz, all-time series are filtered by a low-pass Chebyshev type II digital filter. In addition, an artificial noise (5% of amplitude) is added to the numerical signal for a more realistic representation of the measured rail response. An example of a comparison of measured responses regarding the passage of a freight train considering a healthy wheel and a wheel with a semi-imperfection profile is shown in Figure 11a. The wheel with a semi-imperfection profile is located on the first wheel of the first wagon. This figure illustrates the similarity between the measured accelerations in each case due to the similar amplitude of healthy and semi-imperfection wheels.

Acceleration time series at position 1 for damaged wheels are presented in Figure 11b, considering two different amplitudes of wheel irregularity profiles. By comparing the results from Figure 11a,b, it is evident that the acceleration variation for the healthy wheel is between −1.5 to 2.5 m/s^2^, while the defective one changes between −70 to 100 m/s^2^. 

## 5. Automatic Wheel Polygonization Detection

### 5.1. Feature Extraction—ARX vs. CWT

As the first step in the automatic damage detection method, damage-sensitive features are extracted from dynamic responses. As mentioned in Section 2.1, the ARX and CWT models are implemented and their performances as feature extractors are compared.

To establish a suitable ARX model order, AIC values are ascendingly ranked from 1 to 70. An average AIC function is presented in Figure 12 based on 113 baseline analyses. It is evident from this figure that after model order 40, the AIC values tend to stabilize, showing that higher orders do not provide additional information. Therefore, the ARX models’ input and output orders are set to 40, and a total of 80 features are extracted. The sensor installed at position 5 ($y_{j}$ in Equation (1)) is used as output, while each of the other sensors considered $(x_{j}$ in Equation (1)) are defined as inputs.

For every 8 accelerometers located on the rail (4 on the right side and 4 on the left side, as shown in Figure 10), 143 baseline scenarios (including 113 simulations for healthy wheels and 30 simulations for semi-imperfection wheels), and 30 damaged scenarios, a total of 468 and 80 features are extracted by the CWT and the ARX methods, respectively. As a result, 3-dimensional feature matrices of *173-by-468-by-8* for CWT and *173-by-80-by-8* (*m-by-f-by-i*) for the ARX models are obtained. 

Figure 13 presents four features for accelerometer 1 (two wavelet coefficients and two ARX parameters) from the first step of the methodology. In this figure, various trends can be observed in the analyzed data for each feature. Moreover, the amplitude dispersion of the features extracted by the ARX models is higher than the CWT method. Figure 13a illustrates how the amplitude variation of the features is less affected by wheel defects (damaged scenarios) and more by EOVs (baseline scenarios). Aside from this, the figure demonstrates the sensitivity of features to baseline scenarios because of variations in speed and irregularities. In Figure 13b,c, a comparison of baseline and damaged scenarios shows similar variations in amplitude. In Figure 13d, the amplitude variation between healthy and damaged scenarios is evident, as the amplitude does not change much for the first 143 passages, and then increases, which may indicate a wheel defect. Overall, the presence of EOVs makes it challenging to distinguish between baseline and damaged scenarios regardless of the technique applied. Thus, the feature normalization step is implemented in the next step. 

### 5.2. Feature Normalization–PCA

The normalization of data related to environmental and operational effects is carried out to obtain indicators that are more sensitive to damage and less sensitive to EOVs. The Latent Variable Method PCA is applied for data normalization. As explained in Section 5.1, for all 173 train passages, 3-dimensional CWT-double-PCA-based feature matrices of 173 × 468 are calculated for each sensor. At the same time, the ARX model is implemented to extract a feature matrix of 173 × 80. During the current step, a PCA-based model is applied to the parameters and new matrices of normalized features are achieved for each sensor. Note that the PCA matrices have the same dimensions as the feature extraction step using either CWT or ARX models.

In Figure 14, the same four features shown in Figure 13 for sensor 1 are shown after implementing PCA for all undamaged and damaged scenarios. During the modeling process, components with a cumulative percentage of variances higher than 80% are discarded. Therefore, the number of discarded rows is seventeen and two for CWT-double-PCA-based and ARX-PCA-based methods, respectively. The results obtained are quite different depending on the method applied as a feature extractor. A reduction in the dispersion of the amplitude of the baseline ARX-based features after applying PCA can be observed in Figure 14a,b. However, CWT-double-PCA-based parameters in Figure 14c,d, show a more effective normalization of the baseline features. Overall, both methods reduce operational effects without affecting sensitivity to damage.

### 5.3. Data Fusion—Mahalanobis Distance

To merge all the data obtained from the ARX and CWT methods, the Mahalanobis distance is implemented by applying a two-stage fusion process: in the first step, the features from each sensor are merged and, in the second stage, the multi-sensor information is fused to enhance the sensibility to the damage. Therefore, in the first stage, Mahalanobis distance transfers 468 CWT-double-PCA-based parameters into one column (173 × 1) for each sensor and train passage. At the same time, 80 ARX-PCA-based parameters are converted to a single damage-sensitive feature (173 × 1). Figure 15 demonstrates the DI values for all 173 train passages considering the response from different sensors for ARX-PCA and CWT-double-PCA considering sensor 1. It is possible to observe that this process significantly improves damage detection sensitivity for both approaches.

Afterwards, in the second stage, all sensor information is merged to improve the sensitivity to the damage even more. According to Figure 16, this step allows a clear distinction between damaged and undamaged scenarios for both approaches (ARX-PCA and CWT-double-PCA). It significantly increases the possibility of effective automatic damage detection. 

### 5.4. Automatic Detection—Outlier Analysis

Despite the visual distinction between healthy and damaged wheels using data processing discussed above, strategies for unsupervised wayside monitoring should employ machine learning algorithms capable of recognizing behaviors related to damaged and healthy states [15,16,22,35]. Moreover, comparing the ARX and CWT methods is considered to find the best feature extraction technique in the machine learning strategy that leads to the most reliable results. The last step of the polygonized wheel detection technique is to construct a confidence boundary (CB) using the Gaussian inverse cumulative distribution function. The significance level of the threshold is defined as 1%, as it has been commonly observed in several works addressing damage detection [15,61].

By comparing the confidence boundary (143 passages) with different damage indexes (173 passages), Figure 17 illustrates the effectiveness of the proposed methodology for distinguishing undamaged from damaged scenarios. The results are evaluated for a set of eight sensors (four on each side) located on the rail at mid-span between the sleepers. They show that the methodology can successfully detect all damage scenarios without any false positives or negatives using the CWT technique for feature extraction (Figure 17b). On the other hand, when the input signal is evaluated by the ARX method, two false positives are observed. Therefore, it is possible to conclude that using ARX models as feature extractors leads to a slightly less efficient polygonized wheel detection.

## 6. Sensitivity Analysis

To reduce the time of installation and maintenance costs, it is necessary to obtain an optimal number of sensors, as well as a suitable location to install them without compromising the quality of the detection. As a result, a parametric study is carried out using accelerometers at distinct locations. In this analysis, only CWT is used for feature extraction because, as shown in Figure 17b, it is the best technique (with no false positives) compared to the ARX method, which leads to two false positives.

### 6.1. Sensitivity Analysis to The Number of Sensors

Several configurations of accelerometers located on the rail between two sleepers (as presented in Figure 10) in terms of the total number (two, four, and six) are analyzed and the effectiveness of the detection methodology is demonstrated in Figure 18. The results concluded that even with two sensors (one on each side of the rail), the proposed methodology could successfully detect all damage scenarios without any false positives or negatives.

### 6.2. Sensitivity Analysis to the Location of Sensors

In the previous section, the proposed approach to detect a defective wheel from a healthy one was implemented for accelerometers located on the rail. This section validates the methodology for accelerometers placed on the sleeper. Figure 19 presents results from applying the automatic wheel polygonized detection with the CWT technique for feature extraction considering eight accelerometers installed on the sleepers.

The robustness of the proposed method to detect a defective wheel with fewer sensors installed on the sleepers without compromising data quality is discussed in this section and the results are shown in Table 4. For a minimal layout with two sensors, the methodology is proven to be effective in detecting the damage with zero false positives or negatives when the sensors are installed on the rail. Likewise, when the sensors are in the sleepers, the results are also very good, presenting only one false positive and zero false negatives. It is concluded that the proposed methodology has an outstanding sensitivity for both sensor locations, on the rail and on the sleepers.

## 7. Conclusions

A methodology for detecting unsupervised damage in a train wheel is presented in this paper to automatically detect a polygonized wheel from a healthy one. The proposed methodology includes the process of fusing sets of acceleration evaluated on the rail/sleeper to improve sensitivity and involves: (i) acquisition of data from installed sensors; (ii) feature extraction from acquired responses; (iii) feature normalization to suppress environmental and operational effects; (iv) data fusion to merge features without losing information of wheel defect; and (v) classification of the extracted features into two subcategories: healthy wheels or defective ones. As a result of this study, the following main achievements have been made:i.the proposed methodology is robust and cost-effective for detecting wheel defects under real-world conditions. A healthy wheel can be distinguished from a polygonized one, regardless of the rail irregularities, train speed, and train loading;ii.the methodology can successfully detect all damage scenarios without any false positives or negatives when the input signals are evaluated using a CWT technique as a feature extraction; when considering the ARX model in the feature extraction, some false positive detections can be observed, which allows concluding that the use of this technique leads to a slightly less efficient polygonized wheel detection;iii.the system is always able to detect polygonized wheels despite the location of sensors in the track (rail or sleeper), which is a significant advantage regarding the installation process;iv.the proposed methodology offers advantages over previous ones [57,59], as only two sensors are sufficient to detect a polygonized wheel, reducing installation costs and allowing a simpler and more straightforward implementation.

In summary, the achieved results demonstrate the high potential of this innovative application of data mining in the railway industry, especially for infrastructure managers.

As for future developments, the proposed methodology will be validated through field trials based on on-site measurements. Furthermore, the robustness and efficiency of the methodology under distinct track environments, particularly in the presence of bridges, tunnels and other under passing structures, will be evaluated.

## Figures and Tables

**Figure 1 sensors-23-02188-f001:**
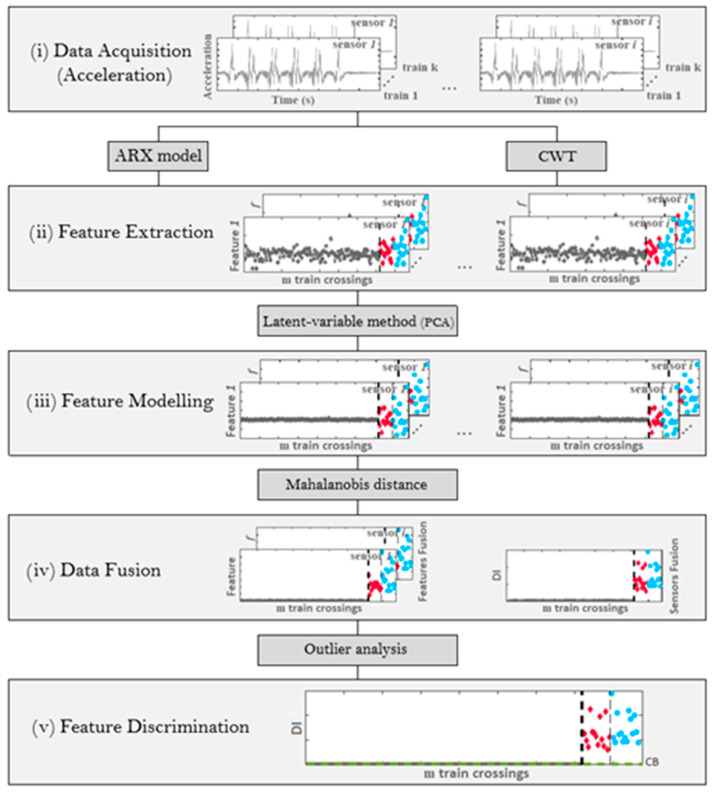
Flowchart of the damage detection methodology.

**Figure 2 sensors-23-02188-f002:**
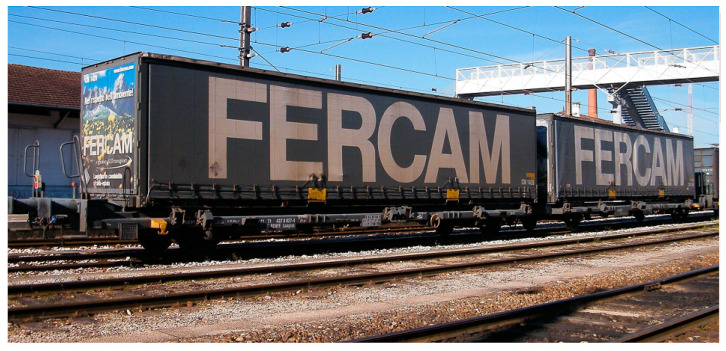
Wagon overview.

**Figure 3 sensors-23-02188-f003:**
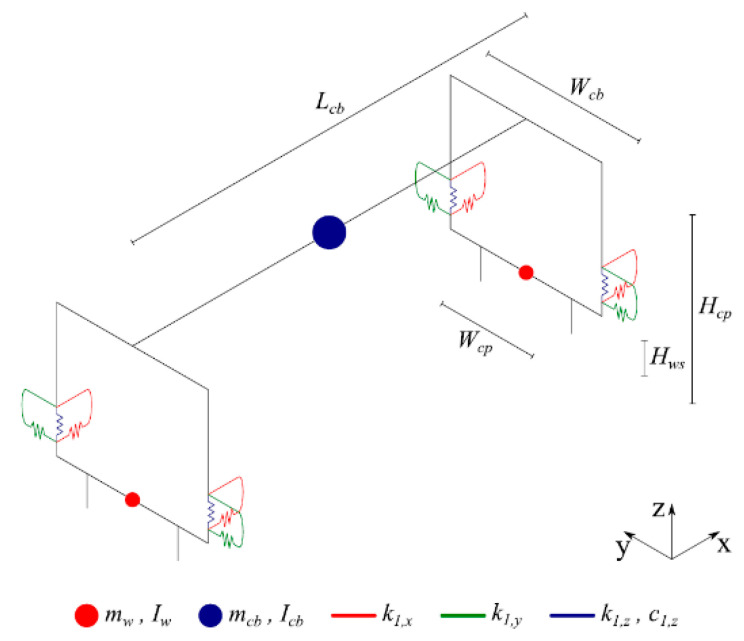
Vehicle numerical model.

**Figure 4 sensors-23-02188-f004:**
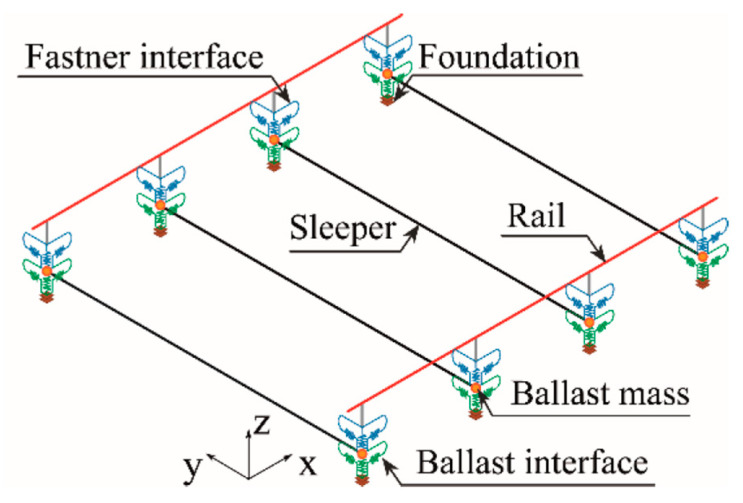
3D FE model of the track.

**Figure 5 sensors-23-02188-f005:**
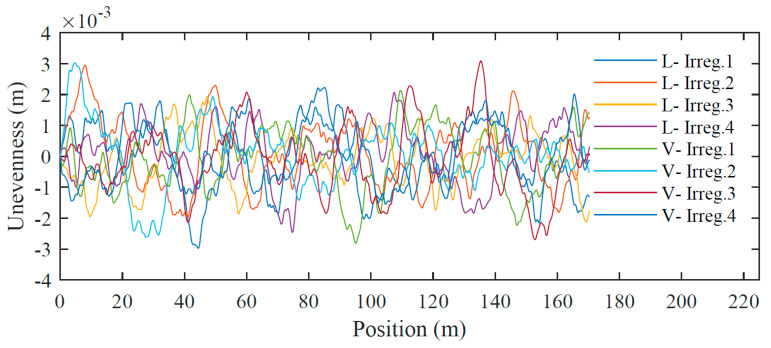
Track irregularity profiles.

**Figure 6 sensors-23-02188-f006:**
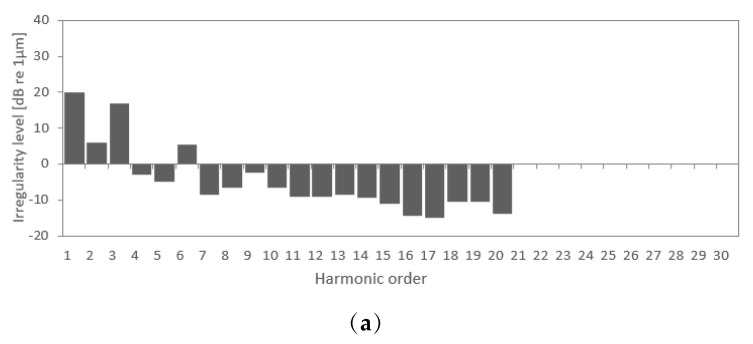
Wheel irregularity amplitude spectra (*Lw*) and harmonic order (*θ*) (**a**) initial polygonal wheel (Johansson et al. [54]), (**b**) damaged polygonal wheel (Cai et al. [55]).

**Figure 7 sensors-23-02188-f007:**
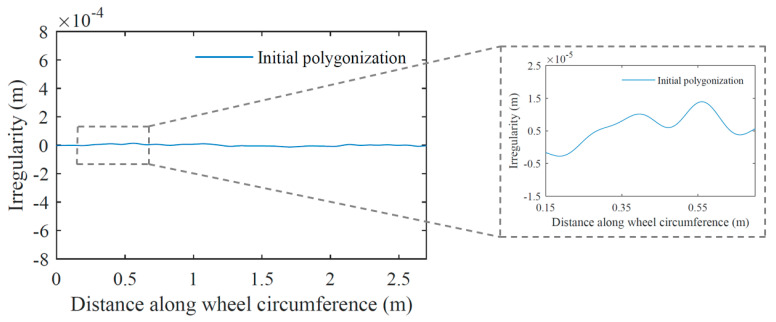
Example of an initial polygonization profile based on Figure 6a irregularity spectrum.

**Figure 8 sensors-23-02188-f008:**
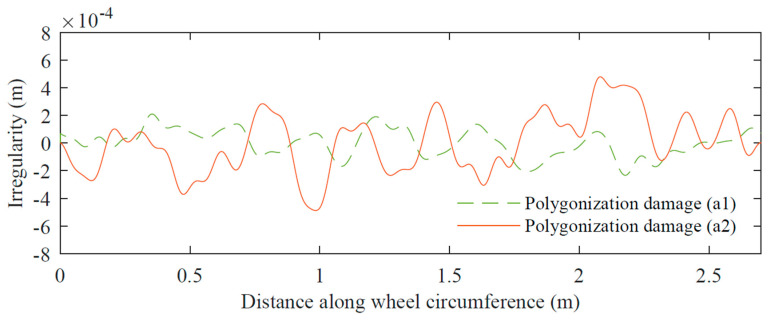
Examples of polygonization-damaged profiles based on Figure 6b irregularity spectrum with different wear ranges: a1 (0.2–0.6) mm and a2 (0.8–1.2) mm.

**Figure 9 sensors-23-02188-f009:**
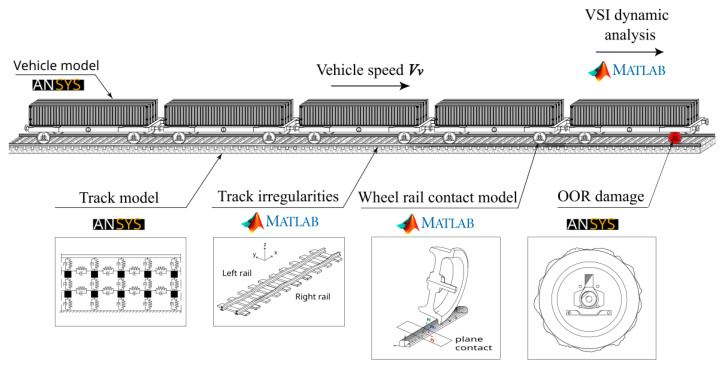
Graphical representation of the numerical modeling.

**Figure 10 sensors-23-02188-f010:**
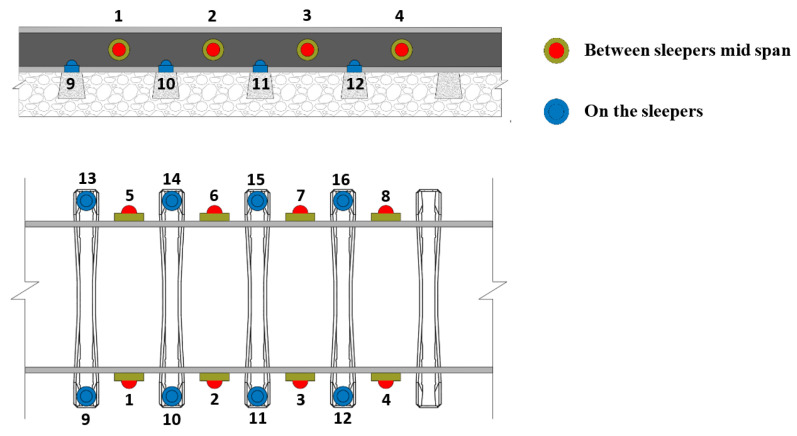
Virtual wayside monitoring system.

**Figure 11 sensors-23-02188-f011:**
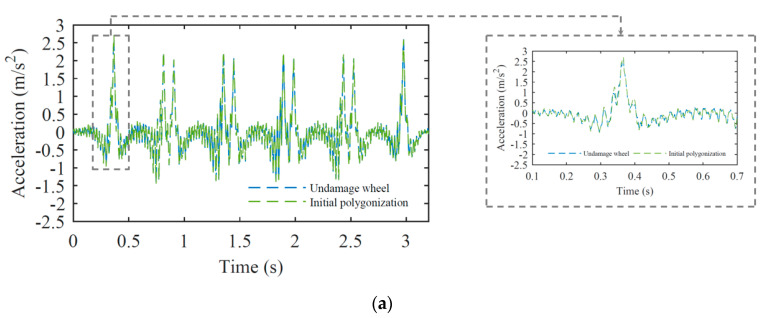
Acceleration time-series on position 1 for (**a**) healthy wheel and wheel with semi-imperfection profile, (**b**) damaged wheel.

**Figure 12 sensors-23-02188-f012:**
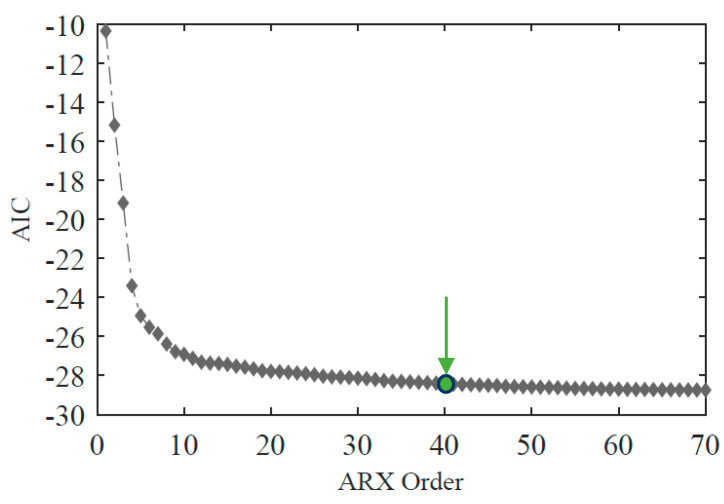
ARX model order.

**Figure 13 sensors-23-02188-f013:**
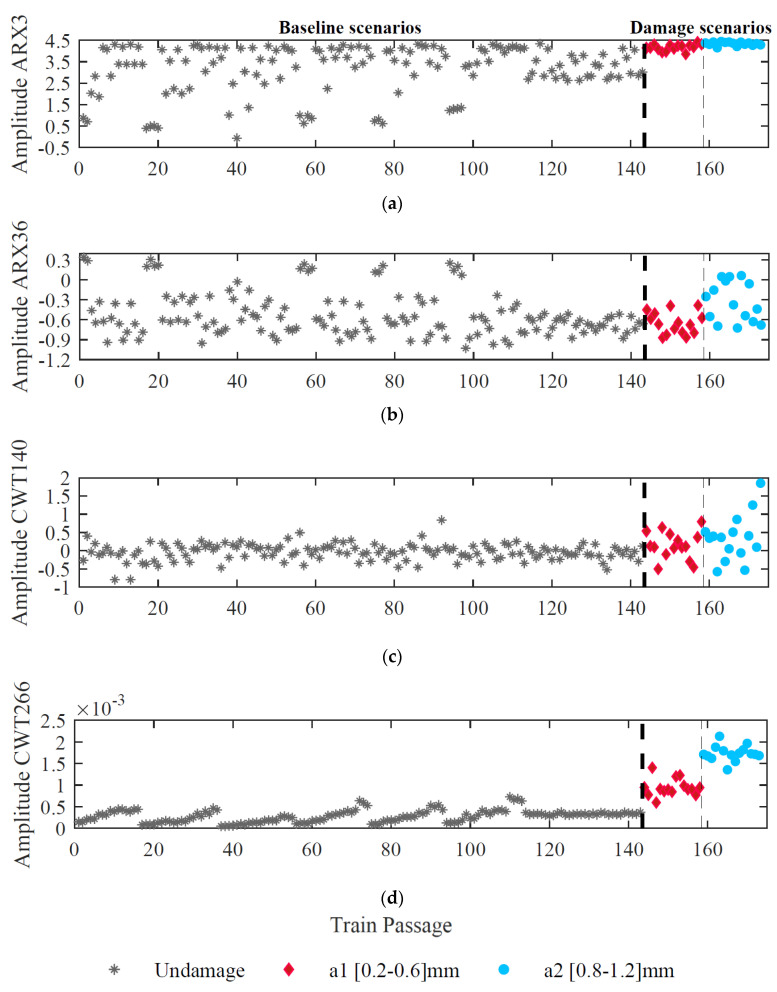
Feature extraction for sensor 1, (**a**) ARX–feature 3, (**b**) ARX–feature 36, (**c**) CWT–feature 140, (**d**) CWT–feature 266.

**Figure 14 sensors-23-02188-f014:**
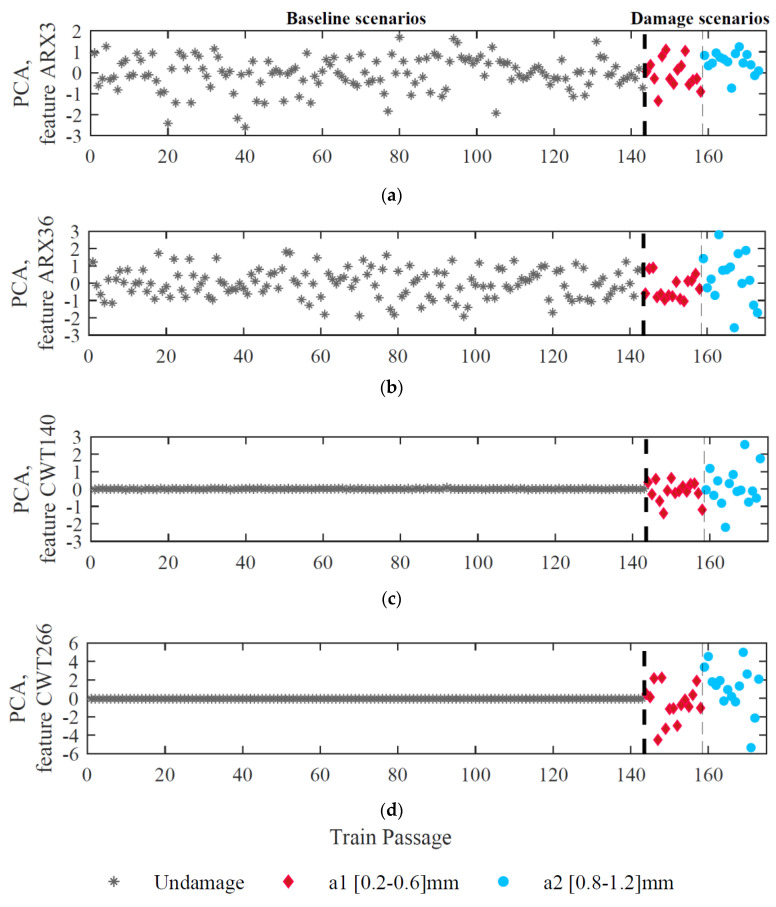
Feature normalization for sensor 1, (**a**) ARX–PCA–based feature 3, (**b**) ARX–PCA–based feature 36, (**c**) CWT–double–PCA-based feature 140, (**d**) CWT–double–PCA–based feature 266.

**Figure 15 sensors-23-02188-f015:**
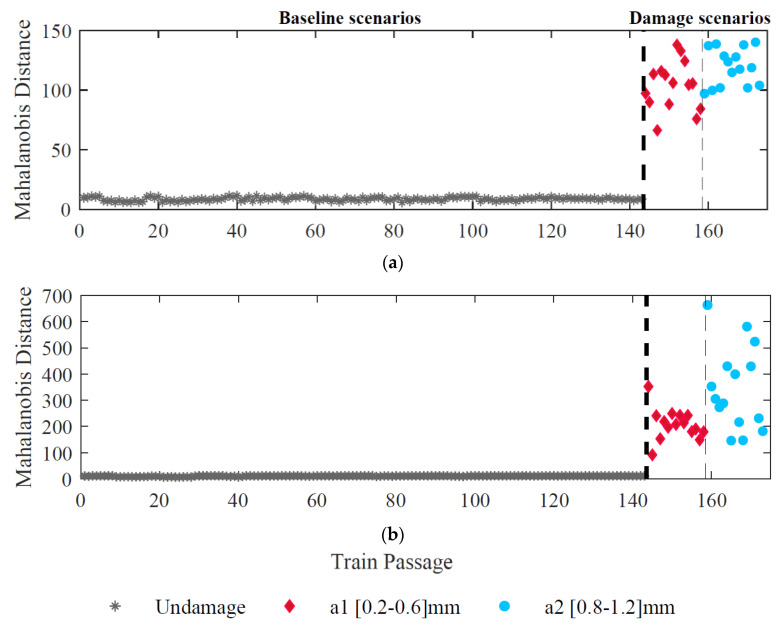
Features fusion for sensor 1, (**a**) ARX–PCA–based, (**b**) CWT–double–PCA–based.

**Figure 16 sensors-23-02188-f016:**
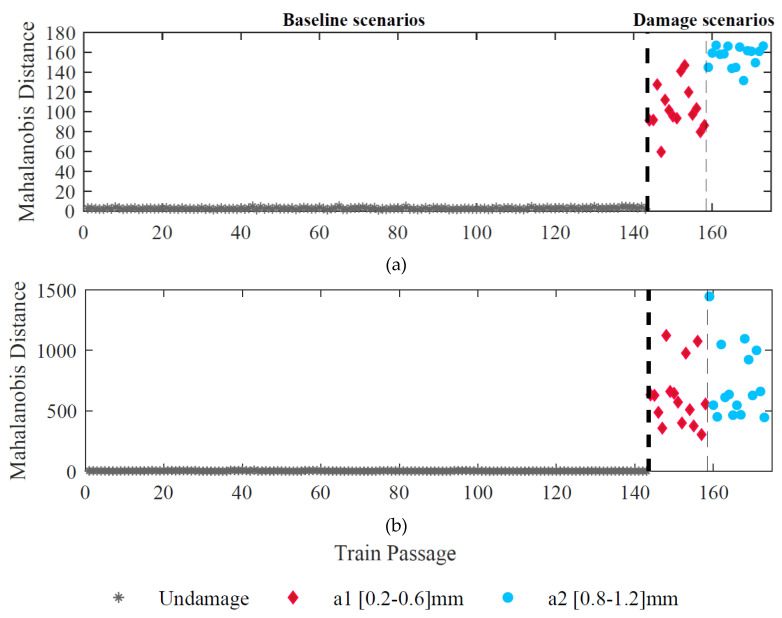
Sensors fusion, (**a**) ARX–PCA–based, (**b**) CWT–double–PCA–based.

**Figure 17 sensors-23-02188-f017:**
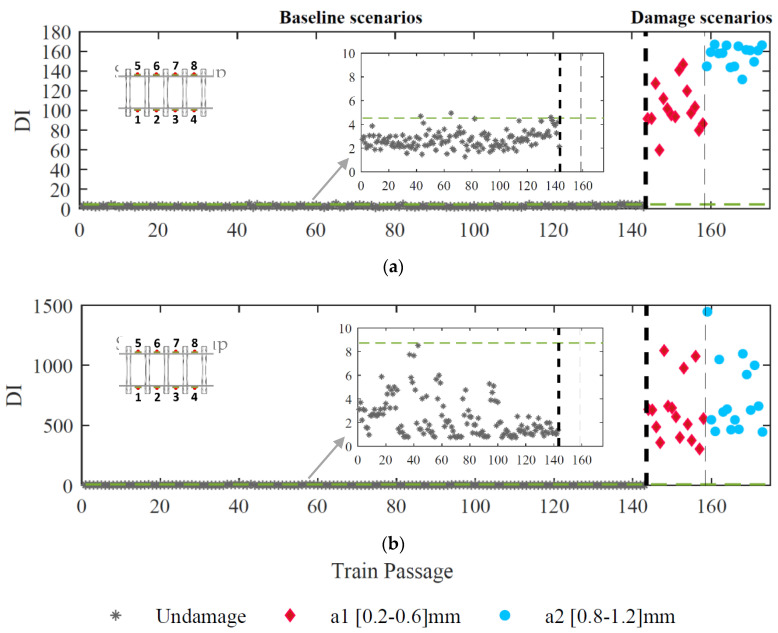
Automatic wheel defect detection (**a**) ARX–PCA–based (**b**) CWT–double–PCA–based.

**Figure 18 sensors-23-02188-f018:**
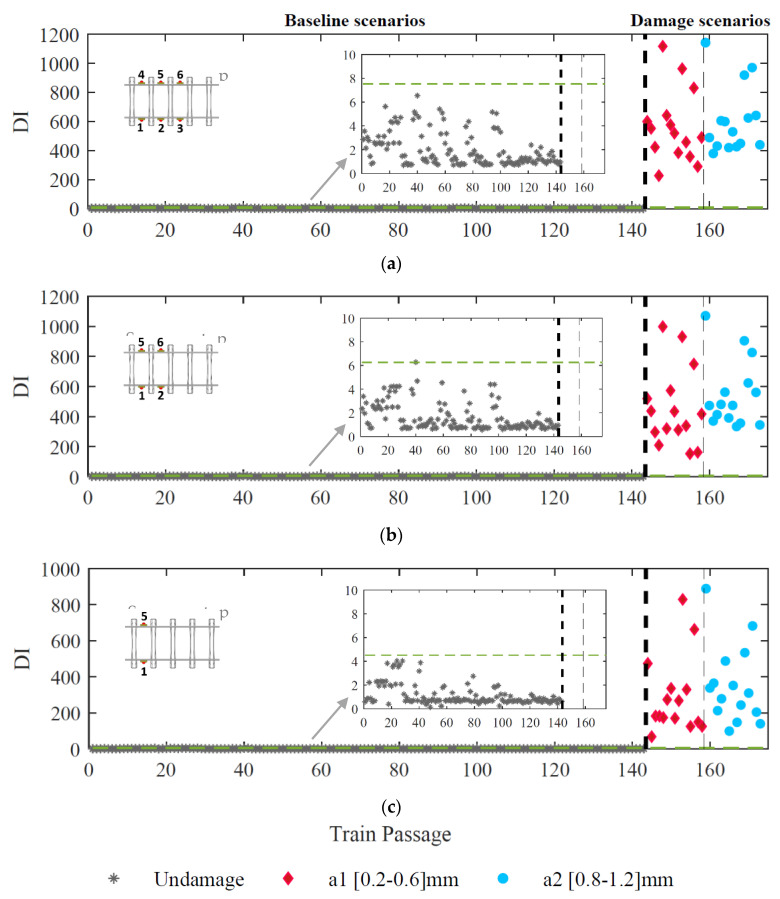
Automatic polygonized wheel detection based on CWT technique considering: (**a**) six sensors, (**b**) four sensors, (**c**) two sensors.

**Figure 19 sensors-23-02188-f019:**
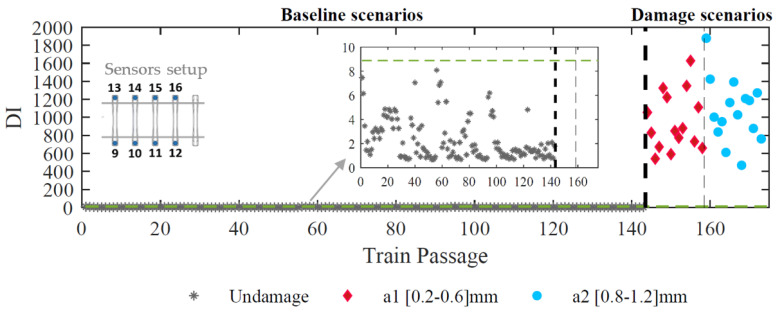
Automatic polygonized wheel detection based on CWT technique considering eight sensors installed on the sleepers.

**Table 1 sensors-23-02188-t001:** Vehicle mechanical and geometric properties.

Parameter	Symbol (Unit)	Adopted Value
**Carbody**		
Mass	$m_{\mathrm{cb}}\;\left(t\right)$	41.1
Roll moment of inertia	$I_{\mathrm{cb},x}\;\left(t.m^{2}\right)$	49
Pitch moment of inertia	$I_{\mathrm{cb},y}\;\left(t.m^{2}\right)$	673
Yaw moment of inertia	$I_{\mathrm{cb},z}\;\left(t.m^{2}\right)$	665
Length	$L_{\mathrm{cb}}\;\left(m\right)$	10,000
**Wheelset**		
Mass	$m_{w}\;\left(\mathrm{kg}\right)$	1247
Roll moment of inertia	$I_{w,x}\;\left(\mathrm{kg}.m^{2}\right)$	312
Yaw moment of inertia	$I_{w,z}\;\left(\mathrm{kg}.m^{2}\right)$	312
**Suspensions**		
Longitudinal stiffness	$k_{1,x}\;\left(\mathrm{kN}/m\right)$	44,981
Lateral stiffness	$k_{1,y}\;\left(\mathrm{kN}/m\right)$	30,948
Vertical stiffness	$k_{1,z}\;\left(\mathrm{kN}/m\right)$	1860
Vertical damping	$c_{1,z}\;\left(\mathrm{kN}.s/m\right)$	16.7

**Table 2 sensors-23-02188-t002:** Mechanical properties of the track.

Parameter	Symbol (Unit)	Value
Rail	$A_{r}\;\left(m^{2}\right)$	$7.67\times 10^{-4}$
	$\rho_{r}\;\left(\mathrm{kg}.m^{3}\right)$	7850
	$I_{r}\;\left(m^{4}\right)$	$30.38\times 10^{-6}$
	$E_{r}\;\left(N/m^{2}\right)$	$210\times 10^{9}$
Rail pad, longitudinal	$k_{p,x}\;\left(N/\;m\right)\;C_{p,x}\;\left(N.s/m\right)$	$20\times 10^{6}\;50\times 10^{3}$
Rail pad, lateral	$k_{p,y}\;\left(N/\;m\right)\;C_{p,y}\;\left(N.s/m\right)$	$20\times 10^{6}\;50\times 10^{3}$
Rail pad, vertical	$k_{p,z}\;\left(N/\;m\right)\;C_{p,z}\left(N.s/m\right)$	$500\times 10^{6}\;200\times 10^{3}$
Sleeper	$\rho_{s}\;\left(N/\;m\right)$	2590
Ballast, longitudinal	$k_{b,x}\;\left(N/\;m\right)\;C_{b,x}\;\left(N.s/m\right)$	$900\times 10^{3}\;15\times 10^{3}$
Ballast, lateral	$k_{b,y}\;\left(N/\;m\right)\;C_{b,y}\;\left(N.s/m\right)$	$2250\times 10^{3}\;15\times 10^{3}$
Ballast, vertical	$k_{b,z}\;\left(N/\;m\right)\;C_{b,z}\;\left(N.s/m\right)$	$30\times 10^{6}\;15\times 10^{3}$
Foundation, longitudinal	$k_{f,x}\;\left(N/\;m\right)$	$20\times 10^{6}$
Foundation, lateral	$k_{f,y}\;\left(N/\;m\right)$	$20\times 10^{6}$
Foundation, vertical	$k_{f,z}\;\left(N/\;m\right)$	$20\times 10^{6}$

**Table 3 sensors-23-02188-t003:** Damaged and undamaged scenarios.

	Undamaged Scenarios/Healthy Wheel		Damaged Scenarios
	Perfect Wheel	Initial Polygonal Wheel	Polygonization Wheel
Vehicle	five freight wagons of the Laagrss type		
Number of loading schemes	6	1 (full load)	1 (full load)
Unevenness profile	4	1	1
Speed range	40–120 km/h	80 km/h	80 km/h
Noise ratio	5%	5%	5%
Location of defect		1st wagon 3rd wagon 5th wagon	1st wagon
Amplitude of defect (W)	-	0.035 mm	0.2–0.6 mm 0.8–1.2 mm
**Total analyses**	**113**	**30**	**30**

**Table 4 sensors-23-02188-t004:** False detections concerning the number of sensors installed in the rail/sleepers with the CWT technique.

Location	Number of Sensors	False Positives	False Negatives
Rail	8 (4 each sides)	0/143 = 0%	0/30 = 0%
	6 (3 each sides)	0/143 = 0%	0/30 = 0%
	4 (2 each sides)	1/143 = 0.7%	0/30 = 0%
	2 (1 each sides)	0/143 = 0%	0/30 = 0%
Sleepers	8 (4 each sides)	0/143 = 0%	0/30 = 0%
	6 (3 each sides)	0/143 = 0%	0/30 = 0%
	4 (2 each sides)	1/143 = 0.7%	0/30 = 0%
	2 (1 each sides)	1/143 = 0.7%	0/30 = 0%

## Data Availability

Not applicable.

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
