# Peer review of "Detection of Wheel Polygonization Based on Wayside Monitoring and Artificial Intelligence"

_sensors, 2023, doi:10.3390/s23042188_

Round 1

Reviewer 1 Report

The manuscript mainly uses artificial intelligence technology to detect wheel polygon problems. First, the manuscript designs a train-track interaction model and simulates the sensors next to the track that the train passes. The feature extraction performance of autoregressive exogenous model and continuous wavelet transform is compared. This research is necessary and has high significance. The manuscript has undergone detailed literature research and has a rigorous structure and logic. However, the manuscript has some issues that need to be further discussed:

1. The author chose roadside sensors in order to reduce the cost of installing sensors. However, does the author consider the difference of the model in different environments like bridge/cave/plateau and its effect on the train track interaction?

2. What is the relationship between where the sensor is installed and "identifying a defective wheel at an early stage"?

3. In Line 27/253/260, there is an error.

4. The author uses several algorithms to process the data. How to avoid the shortcomings of each algorithm and improve the accuracy of the results?

Reviewer 2 Report

The manuscript deals with AI based wheel defect detection approach, wherein the dynamic response is attained considering the application of a train-track interaction model that simulates the passage of the train over a set of accelerometers installed on the rail and sleepers.

The manuscript is written well. However, some observation and comments are as follows:

# Why the research talks about wheel polygonization, not the wheel condition in general or wheel flat defect? General studies say that a wheel polygonization defect is considered a minor wheel defect.

# Discussion of utilized sensors and their mounting location needs to be improved. 

# Any specific reason to utilize autoregressive models (ARX) and Continuous wavelet transform (CWT) feature extraction methods?

Reviewer 3 Report

Paper is well written and explained. Howeverauthor should avoid lumped references andexplaineach reference seperately. Alsoexplain each figure and table.

Reviewer 4 Report

1) Please provide some quantitative explanation in the result part.

2) Please summarize the literature at the end of each paragraph in introduction section as appropriate, which would avoid making your literature review like stacking papers and also help the audience a better view of your review work and better transaction to the next paragraph.

3) Some attention needs to be given to the limitations associated with the methods used in the current work.
